# Management of Brain and Leptomeningeal Metastases from Breast Cancer

**DOI:** 10.3390/ijms21228534

**Published:** 2020-11-12

**Authors:** Alessia Pellerino, Valeria Internò, Francesca Mo, Federica Franchino, Riccardo Soffietti, Roberta Rudà

**Affiliations:** 1Department of Neuro-Oncology, University and City of Health and Science Hospital, 10126 Turin, Italy; francesca.mo374@edu.unito.it (F.M.); fedef8@virgilio.it (F.F.); riccardo.soffietti@unito.it (R.S.); rudarob@hotmail.com (R.R.); 2Department of Biomedical Sciences and Human Oncology, University of Bari Aldo Moro, 70121 Bari, Italy; valeria.interno@libero.it; 3Department of Neurology, Castelfranco Veneto and Treviso Hospital, 31100 Treviso, Italy

**Keywords:** brain metastases, leptomeningeal metastases, HER2-positive breast cancer, ER-positive breast cancer, triple negative breast cancer, clinical trials, targeted agents, selection criteria

## Abstract

The management of breast cancer (BC) has rapidly evolved in the last 20 years. The improvement of systemic therapy allows a remarkable control of extracranial disease. However, brain (BM) and leptomeningeal metastases (LM) are frequent complications of advanced BC and represent a challenging issue for clinicians. Some prognostic scales designed for metastatic BC have been employed to select fit patients for adequate therapy and enrollment in clinical trials. Different systemic drugs, such as targeted therapies with either monoclonal antibodies or small tyrosine kinase molecules, or modified chemotherapeutic agents are under investigation. Major aims are to improve the penetration of active drugs through the blood–brain barrier (BBB) or brain–tumor barrier (BTB), and establish the best sequence and timing of radiotherapy and systemic therapy to avoid neurocognitive impairment. Moreover, pharmacologic prevention is a new concept driven by the efficacy of targeted agents on macrometastases from specific molecular subgroups. This review aims to provide an overview of the clinical and molecular factors involved in the selection of patients for local and/or systemic therapy, as well as the results of clinical trials on advanced BC. Moreover, insight on promising therapeutic options and potential directions of future therapeutic targets against BBB and microenvironment are discussed.

## 1. Introduction

Breast cancer (BC) is the second most common solid tumor that can metastasize to CNS. Approximately 30% of patients develop brain metastases (BM) [1], and other 5% leptomeningeal metastases (LM) [2]. The prognosis of patients with CNS involvement ranges from months to years based on different clinical and molecular factors. A retrospective cohort of 423 patients with BM/LM from BC, analyzed between 2005 and 2015, reported a median overall survival (OS) of 6.9 months (95% CI 5.5–7.8), and one- and two-year OS of 35% and 17%, respectively [3]. Karnofsky Performance Status (KPS) > 70, single BM, and absence of LM or extracranial disease confer a prolonged OS. BC were classified based on the 2015 ESMO Guidelines [4] in five molecular subtypes: luminal A (estrogen receptor/ER+, HER2-negative, low ki67, high PR), luminal B human epidermal growth factor receptor 2 (HER2)-negative (ER+, HER2-, high ki67, or low PR+), luminal B HER2-positive (ER+, HER2+, any PR, any ki67), HER2-enriched (HER2+, ER-, PR-), and “basal like” or triple negative BC (ER-, PR-, HER2-negative/TNBC). The analysis showed that the molecular subtypes were significantly correlated with OS (*p* < 0.0001): 3.1 months (95% CI 2.4–3.9) for TNBC, 3.9 months (95% CI 2.3–5.6) for luminal B HER2-negative, 7.1 months (95% CI 4.3–9.8) for luminal A, 12.1 months (95% CI 8.3–15.9) for HER2-enriched, and 15.4 months (95% CI 8.8–22.1) for luminal B HER2-positive, respectively [3]. Other studies reported a median OS for patients with luminal B and HER2-positive BC of 7.1–18.9 months and 13.1–16.5 months, respectively, while 4.4–4.9 months for TNBC patients [5,6]. In general, BM occur in 8–15% and 11% of luminal A and B advanced BC, and in 11–48% and 25–46% of HER2-enriched BC and TNBC, respectively [5,7]. Approximately 43% of patients with BM may develop secondary LM [8]. Patients with LM from BC have a poor prognosis with a median OS of 2.0 months (95% CI 0.1–4.3). However, HER2-enriched LM treated with targeted therapy displayed a prolonged survival (11.4 months for HER2+/ER-, and 6.6 months for HER2+/ER+, respectively). Among clinical factors, Eastern Cooperative Oncology Group Performance Status (ECOG PS) > 2 had a significantly shorter median OS than patients with ECOG PS ≤ 2 (HR 2.35; 95%CI 1.64–3.37. Moreover, patients with ≥ 2 WBC (white blood cells)/mm^3^ (HR 3.4; 95%CI 1.8–5.0), glucose levels ≥ 3 mmol/L (HR 7.4, 95%CI 4.7–10.0), and protein levels ≥1 g/L (HR 2.4; 95%CI 0.6–4.3) in cerebrospinal fluid (CSF) had a significantly shorter median OS [8]. Another cohort of 187 LM from BC treated from 1999 to 2015 showed a median OS of 4.2 months with a 6- and 12-months OS of 34 and 15%, respectively [9]. Factors that positively impact the OS were age ≥ 53 years (HR 1.63; 95%CI 0.18–2.25; *p* = 0.003) KPS ≥ 70 (HR 0.61, 95%CI 0.43–0.88; *p* = 0.008), luminal A and B subtypes (HR 0.64; 95%CI 0.46–0.88; *p* = 0.007), systemic treatment (HR 0.41; 95%CI 0.286; 0.611; *p* < 0.001), intrathecal treatment (HR 0.68; 95%CI 0.49–0.96; *p* = 0.029), and radiation therapy (HR 0.47; 95%CI 0.32–0.69; *p* < 0.001).

Overall, the frequency of BM and LM is increasing as treatment of primary BC and imaging techniques have improved [10]. Another reason for the rising incidence of CNS involvement following systemic treatments is that most of targeted agents and traditional chemotherapy have poor penetration through the blood-brain barrier (BBB) [11]. In this regard, trastuzumab represents an example of a high molecular-weight molecule with a significant efficacy to control systemic disease, but with poor penetration through the normal BBB resulting in a limited intracranial disease control. Olson et al. performed a meta-analysis on 9020 patients with HER2-positive BC treated or not with adjuvant trastuzumab for one year. The incidence of BM as a first site of recurrence in HER2-positive patients receiving adjuvant trastuzumab was higher (2.56%; 95%CI 2.07–3.01) compared with those who did not receive trastuzumab (1.94%; 95%CI 1.54–2.38) with an increased relative risk of 1.35 (95%CI 1.02–1.78, *p* = 0.038) to have a CNS recurrence following adjuvant trastuzumab [12].

## 2. Prognostic Scales in Advanced Breast Cancer with CNS Disease

Some prognostic scales have been developed for helping clinicians to choose the adequate treatment. In 2010, Niwinska et al. used a recursive partitioning analysis in 441 newly diagnosed BM from BC (B-RPA). The B-RPA allowed to identify 3 different prognostic classes: class I included patients with 1–2 BM, without extracranial disease or with controlled extracranial disease, and KPS of 100. Class III included patients with > 2 BM with KPS of ≤60. Class II included all other patients. The median OS was 29.0 for class I, 9.0 for class II, and 2.4 months for class III, respectively (*p* < 0.0001) [13]. Sperduto et al. initially developed a Graded Prognostic Assessment (GPA) for patients with BM [14] with the further addition of molecular subtypes to clinical factors (age and KPS) (Breast-GPA index): the combination of these factors led to 4 prognostic groups with different OS. In particular, group 1 (GPA 0–1) displayed a median OS of 3.4 months, group 2 (GPA 1.5–2.0) an OS of 7.7 months, group 3 (GPA 2.5–3.0) an OS of 15.1 months, and group 4 (GPA 3.5–4.0) an OS of 25.3 months, respectively. Among HER2-negative patients, the presence of ER/PR positive status improved the median OS from 6.4 to 9.7 months, while among HER2-positive patients, the ER/PR positive status improved the median OS from 17.9 to 20.7 months [15]. A further analysis on a larger cohort of 2473 patients with BM from BC confirmed the prognostic value of GPA groups. The median OS for GPA 0–1.0, 1.5–2.0, 2.5–3.0, and 3.5–4.0 was 6, 13, 24, and 36 months, respectively. Moreover, the median OS in HER2-enriched BM increased from 18 to 25 months compared with HER2-negative BC. Overall, the Authors suggested that the Breast-GPA may help clinicians in decision-making and will be useful for stratification in future clinical trials [16]. Rades et al. proposed another prognostic score (Simple Survival Score for BM—SS-BM) in 230 BC patients with BM who received whole-brain radiotherapy (WBRT). The following six prognostic factors were evaluated for association with OS: WBRT schedule (five fractions of 4 Gy versus 10 fractions of 3 Gy), age (≤ 60 vs. ≥ 61 years), KPS (< 70 or ≥ 70), number of BM (1–3 or ≥ 4), extracranial metastases (no or yes), and time between tumor diagnosis and WBRT (< 36 months or ≥ 36 months). Patients were divided into three prognostic groups according to the combination of prognostic factors: group 1 (score 4–7), group 2 (score 9), group 3 (15 points), with a median six-month OS of 1%, 54%, and 75%, respectively (*p* < 0.001) [17]. Griguolo et al. have proposed a refinement of the Breast-GPA, that comprises number of BM in addition to age, tumour subtype and KPS (Modified Breast-GPA—MB-GPA). The MB-GPA was validated in a multicentric European cohort of 668 BC patients with BM. The number of BM (1–3 and >3) was significantly correlated with OS in univariate analysis (*p* < 0.001), and >3 BM was identified as a negative prognostic factor in multivariate analysis. Although both Breast-GPA and Modified Breast-GPA (MB-GPA) accurately predicted OS (*p* < 0.001), the MB-GPA was more accurate: in fact, concordance indices were 0.639 (95% CI 0.638–0.639) and 0.665 (95% CI 0.664–0.666) for Breast-GPA and MB-GPA, respectively (*p* < 0.001) [18]. Znidaric et al. have retrospectively analyzed 423 BC patients with BM/LM treated with radiation therapy to validate the applicability of 4 different prognostic scales (B-RPA, Breast-GPA, SS-BM, and MB-GPA). SS-BM and MB-GPA showed the best discriminating ability (concordance index of 0.768 and 0.738, respectively) [3]. However, poor KPS (<60), high number of BM (≥3), active extracranial disease, molecular subtypes (TNBC or luminal B), and the presence of LM are considered as risk factors for shorter OS, regardless of the type of prognostic scale [19].

New molecular factors have been suggested to promote metastatic disease in CNS from BC. Xie et al. have found that fibroblast growth factor receptor (FGFR) 1 aberrations tend to occur in ER/PR positive (79/148, 53.4%), and HER2-enriched (30/148, 23.4%) BM from BC, while it is less represented in TNBC (20/148, 15.6%). The combination of FGFR, TP53 and FLT1 aberrations and HER2-positivity were associated with an increased risk of developing BM (AUC 77.13%), and FGFR1 alteration also was a significant risk factor for poor progression-free survival (PFS) (*p* = 0.029) [20]. Overall, molecular factors that influence survival should be integrated in prognostic scales to better stratify patients in clinical practice and future clinical trials.

## 3. Local Treatments for BM and LM from Breast Cancer

Local treatment options for BM from solid tumors are surgery, stereotactic radiosurgery (SRS) and WBRT. Surgery should be performed in patients with a limited number (1–3) of BM, especially in the case of lesions of ≥3 cm in diameter or located in the posterior fossa or eloquent areas. Moreover, the patients should have a KPS ≥ 60 and/or a stable systemic disease and/or effective systemic treatment options [21]. The role of surgery in patients with multiple BM is limited and the most common clinical response is to remove the symptomatic lesion and treat the other ones with SRS. The role of surgery in LM is limited to bulky lesions with mass effect needing an immediate palliation. The National Comprehensive Cancer Network (NCCN) guidelines state that in “limited BM SRS is equally effective and offers significant cognitive protection compared with WBRT” [22]. In general, BM are classified as either limited or extensive disease based on number and volume: one to four BM are considered limited and may be treated with surgery, when feasible, or SRS, while patients with > 4 BM and/or active systemic disease should receive WBRT [23]. There has been a long debate on whether SRS or WBRT is necessary to eradicate microscopic residual disease following surgery. Randomized clinical trials have shown that the omission of WBRT following surgery may lead to a higher risk of CNS relapse while not improving OS. Aoyama et al. have investigated the activity of SRS alone compared with SRS plus WBRT in 132 patients with 1–4 BM (nine from BC) reporting a 12-month BM recurrence rate of 76.4% for SRS alone and of 46.8% in the WBRT + SRS group arm (*p* < 0.001), but not a significant advantage in OS (median OS of 8.0 months and one-year OS of 28.4% for SRS alone, and 7.5 months and 38.5% for the SRS plus WBRT arm, respectively) (*p* = 0.42) [24]. Similarly, Kocher et al. have reported a higher intracranial disease control for SRS plus WBRT (78%) compared with SRS alone (48%) in 359 patients with 1–3 BM (42 from BC), but a similar median OS (10.9 and 10.7 months, respectively) [25]. Last, Brown et al. have shown an intracranial disease control of 84.6% and 50.5% after SRS plus WBRT and SRS alone, respectively, in 213 patients with 1–3 BM (18 from BC), but not a significant difference in median OS (10.4 months and 7.4 months, respectively) [26]. Notably, the use of SRS alone resulted in less cognitive deterioration at three months compared with SRS plus WBRT. Therefore, in the absence of a difference in OS, SRS alone may be a preferred strategy for patients with 1–3 BM [27], while WBRT may be reserved for extended CNS disease as salvage therapy [22,27]. Conversely, according to an old study, the phase III PCI-P120–9801 trial, WBRT improves tumor shrinkage in 208 patients with BM (75 from BC), resulting in a better preservation of neurocognitive functions, especially in long-term survivors (> 15 months from the WBRT) [28].

Some studies have reported a mild improvement of neurocognitive functions in patients who underwent the hippocampal-sparing WBRT [29,30]. Moreover, Gondi et al. have suggested a remarkable benefit in speaking, memory, and patient-reported symptoms, including fatigue and interference in daily living, following WBRT associated with memantine [31]. Westover et al. have investigated, in a single-arm phase II trial, the hippocampal-sparing WBRT with a simultaneous integrated boost (HSIB-WBRT) delivered to BM in 50 patients reporting a median decline in Hopkins Verbal Learning Test-Revised delayed recall (HVLT-R DR) at three months of 10.6%, a cumulative incidence of intracranial recurrence of 8.8%, and a median OS of nine months [32].

The advantage of SRS is to deliver a single fraction of high radiation dose to a well demarcated lesion sparing surrounding normal brain tissue. Another debate is on the role of SRS in the setting of extensive BM. In this regard, the prospective observational study JLGK0901 has enrolled 1194 patients with 1–10 BM (123 from BC) receiving SRS alone. Patients with 5–10 BM had similar OS than those with 2–4 BM (10.8 months (95% CI 9.1–12.7) and 10.8 months (9.4–12.4, respectively)), without any difference in toxicity [33]. Similarly, Hughes et al. retrospectively analyzed 2089 patients with 1–15 BM (295 from BC) treated with SRS alone. The median OS for patients with 2–4 BM (882 patients, 42%) and 5–15 BM (212 patients, 10%) were 9.5 and 7.5 months, respectively, without any significant statistical difference. One-year CNS recurrence was 41% for patients with 2–4 BM, and 50% for those with 5–15 BM [34]. When patients need to receive SRS for multiple BM, the total volume of lesions seems to be more relevant rather than the total number of BM. Molecular subtypes of BM from BC are correlated with different outcomes following SRS. In this regard, the median OS after SRS of BM was longer in luminal/HER2 positive BC (35.8 months), followed by HER2 positive (31.4 months, and shorter OS were observed in luminal B and TNBC (13.7 and 10.4 months, respectively) [35]. This discrepancy in OS between HER2 positive and Luminal patients is due to a longer duration from primary BC diagnosis to BM onset and to the development of chemoresistance of BM to multiple lines of previous chemotherapy in Luminal patients [7]. Moreover, BM from TNBC are associated with a major risk of distant brain recurrence compared with HER2 positive patients, who are more prone to recur locally [35]. Moreover, the prolonged OS observed in BM from HER2 positive patients has been correlated with a better penetration of anti-HER2 targeted therapy, including trastuzumab, under a condition of impaired BBB following radiotherapy [36]. Recently, the American Radium Society’s Appropriate Use Criteria has systematically reviewed the literature on survival and neurocognitive outcomes after SRS in multiple BM. For patients with 2–10 BM, SRS alone is considered as an appropriate treatment option for selected patients with good PS, while is not feasible for patients with >20 BM. There are still several areas of disagreement, including: hippocampal sparing WBRT for 2–4 asymptomatic BM; fractionated- versus single-fraction SRS for resected BM, larger BM and/or brainstem metastases; WBRT versus hippocampal-sparing WBRT versus SRS alone for patients with 11–19 BM, and active systemic disease, poor PS, and no systemic options [37]. Thus, prospective studies are warranted to draw appropriate indications for these scenarios.

Radiotherapy (RT) does not represent the first line treatment in LM for different reasons. First, a retrospective analysis has demonstrated a major impact of systemic chemotherapy and targeted agents in LM control and OS [38]. Moreover, randomized clinical trials, evaluating the efficacy and safety of RT in LM, have not been conducted thus far. Focal RT, such as involved field or SRS, may be considered in patients with local, circumscribed, and symptomatic lesions, or in those with CSF flow obstructions due to spinal or intracranial blocks in order to improve the distribution of intra-CSF therapy. Wolf et al. retrospectively analyzed 16 patients with LM from solid tumor (five from BC), treated with SRS, reporting a disease control of 57.1% (partial response in eight patients) with a median OS of 10 months (six-month and one-year OS of 60% and 26%, respectively) [39]. The Authors suggest that SRS could be added to treat bulky LM in patients also eligible for systemic therapy, including immuno-therapies and targeted therapies, with the aim to prolong OS.

WBRT is not recommended for the treatment of LM because of the poor benefit and the significant risk of developing severe adverse effects (myelotoxicity, enteritis, and mucositis). However, some Authors have investigated the effect of WBRT in unfit patients for systemic treatment and low performance status. In this regard, Gani et al. reported an OS at six- and 12-months of 26% and 15%, respectively, an improvement of neurological deficits in 11%, and a median OS of two months following WBRT in 27 patients with LM from solid tumors (20 from BC) [40]. Brower et al. retrospectively analyzed 124 patients with LM from solid tumors (22 BC) and showed a median OS of 9.2 months when WBRT was associated with systemic chemotherapy, with a major benefit in patients with good KPS (KPS ≤ 50: 1.1 months; KPS 60–80: 2.0 months; KPS 90–100: 5.9 months) [41]. Notably, Brower identified some prognostic factors (KPS ≥ 90 and absence of BM) in patients with prolonged OS as compared with historical controls.

Craniospinal irradiation (CSI) is a more aggressive approach with limited data of efficacy. El Shafie et al. have described 25 patients, who received CSI for LM from solid tumors (15 BC), an reported a neurologic improvement in seven patients (28%), and median OS of 4.8 months (95%CI 2.7–8.0) [42]. Devecka et al. reported in a cohort of 19 patients with LM (five from BC) a median OS of 7.3 months, 3.3 months, and 1.5 months for patients with zero, one, and two risk factors, respectively, according to the proposed prognostic score (KPS < 70 and the presence of extra-CNS disease) [43]. Recently, Yang et al. have investigated in a phase I trial the tolerability of proton CSI in 19 patients with LM from solid tumors (three from BC), reporting a median OS of eight months (95% CI 6 to not reached), of whom four patients (19%) were disease free ≥ 12 months [44] Two patients (10.5%) had grade 4 lymphopenia, grade 4 thrombocytopenia, and grade 3 fatigue. The NCCN 2020 guidelines for management of LM recommend involved-field RT in association with intrathecal chemotherapy in patients with favorable prognostic factors (KPS ≥ 60, mild neurologic deficits, no bulky disease, stable systemic disease, available therapeutic options for systemic disease). For patients who do not meet these criteria, involved-field RT to symptomatic lesions or best supportive care, are the suggested options [22]. The EANO ESMO Guidelines basically overlap the NCCN Guidelines [45].

## 4. Intrathecal Therapy for LM from Breast Cancer

Intrathecal therapy is employed in patients with tumor cells in the CSF and/or with linear diffuse enhancing leptomeningeal disease, while is not effective to treat nodular lesions due to the limited penetration into the tumoral tissue. Three drugs are commonly used: methotrexate (MTX), liposomal cytarabine (Ara-C) and thioTEPA. Five old clinical trials only have investigated the efficacy of intrathecal therapy in LM from solid tumors, including BC. Grossman et al. have investigated the efficacy of intrathecal MTX 10 mg compared with thioTEPA 10 mg twice weekly in a cohort of 52 patients with LM (25 from BC), reporting a median OS for patients receiving MTX of 3.9 months and of 3.5 months for those treated with thioTEPA. No patient had a significant neurologic improvement following intrathecal therapy, and 75% deteriorated neurologically within 8 weeks after the start of treatment [46]. Hitchins et al. evaluated in a prospective randomized trial on intrathecal MTX 15 mg or MTX 15 mg plus Ara-C 50 mg/m2 44 patients with LM (11 from BC) showing an overall response rate of 55% [47]. Response to MTX was superior compared with combined MTX/Ara-C (61% versus 45%, respectively). Seven patients achieved a complete response. Glantz et al. conducted a randomized clinical trial comparing intrathecal Ara-C (31 patients) with MTX (30 patients) reporting a radiological response in 26% of patients treated Ara-C and 20% in those who received MTX (*p* = 0.76). Median OS was similar between the two arms (3.5 months in the Ara-C arm and 2.6 months in the MTX; *p*= 0.15) [48]. Boogerd et al. compared intraventricular chemotherapy (*n* = 17) with non-intrathecal treatment, including systemic chemotherapy and involved-field RT (*n* = 18), in patients with LM from BC: a neurological improvement was observed in 59% of the intrathecal and 67% of non-intrathecal group, with a median PFS of 5.7 months and 6 months, respectively. Median OS of patients receiving intrathecal therapy was of 4.6 months and 7.6 months for patients treated with non-intrathecal therapy (*p* = 0.32) [49]. Last, Le Rhun et al. have investigated the activity of the addition of liposomal Ara-C to systemic therapy in 69 patients with LM from BC. Patients treated with systemic therapy alone achieved a median PFS and OS of 2.0 and 4.0 months, respectively, while those receiving liposomal Ara-C plus systemic therapy reported a median PFS and OS of 4.3 and 7.3 months, respectively [50]. Other compounds have been evaluated for intrathecal treatment, such as trastuzumab for LM from HER-2 enriched BC. A phase I study conducted on 11 patients showed seven stable disease and four progressive disease following administration of intrathecal trastuzumab 150 mg weekly and no serious adverse events were reported [51]. Data of the phase II trial on the efficacy and tolerability of dose-escalated intrathecal trastuzumab are still pending (NCT01325207). Figura et al. have compared the activity of intrathecal trastuzumab (18 patients), intrathecal MTX or thioTEPA (15 patients), or WBRT alone (23 patients) in LM from HER-2 positive BC. Significant differences were found in PFS with six-month rates of 44%, 18%, and 26% (*p* = 0.04) for intrathecal trastuzumab, intrathecal MTX/thioTEPA, and WBRT, respectively [52]. A prolonged disease control > 10 months was achieved in 4 patients treated with intrathecal trastuzumab. Twelve-month OS were 54%, 10%, and 19% (*p* = 0.01) for intrathecal trastuzumab, intrathecal MTX/thioTEPA, and WBRT, respectively. Recently, Zagouri et al. conducted a meta-analysis, that evaluated intrathecal trastuzumab in patients with LM from HER-2 positive BC [53]. Fifty-eight patients were included in the analysis, and intrathecal trastuzumab was used both alone (20 patients) or in combination with systemic chemotherapy (37 patients). A significant clinical improvement was observed in 55.0% of patients, and a stable disease was achieved in 14% of patients. CSF response was observed in 55.6% of patients. MRI was improved or stable in 70.8% of patients. Median PFS was 5.2 months and median OS was 13.2 months following intrathecal trastuzumab. The Authors suggested that intrathecal trastuzumab might be a safe and effective treatment, but further prospective studies are needed with larger cohort for a definitive confirmation. Conversely, when using standard drugs for intrathecal therapy, such as MTX, liposomal Ara-C, and ThioTEPA, relatively old studies reported a limited palliative efficacy in LM from BC with a median OS approximately of 4–6 months. These studies did not analyze the differential impact of intrathecal therapy among different subtypes of BC. A cohort on 153 LM from BC (ER+ 51%, HER2+ 20.3%, and TNBC 15.0%) displayed that TNBC and ER positive patients had significant benefit in OS from intrathecal therapy (HR 0.60, 95%CI 0.37–0.97) and systemic therapy (HR 0.17, 95% CI 0.10–0.29) in multivariable analysis [8]. Overall, the efficacy of intrathecal therapy is still modest, and a careful evaluation of clinical factors helps clinicians to identify the subgroups of patients who may benefit.

## 5. Systemic Targeted-Therapies for BM and LM from BC According to Molecular Subtypes

Typically, CNS recurrence occurs in patients with advanced disease [54], while the risk is low for patients with local BC [12,55]. Although many advancements in systemic therapy have been made, the OS after a CNS relapse still remains limited. The BBB represents the main barrier for the penetration of compounds into the CNS, and different approaches have been developed to better deliver drugs through the BBB, such as design drugs with increased lipophilicity and low molecular weight, use high-dose or pulsatile schedules and modify BBB permeability by chemical or mechanical tools.

### 5.1. HER2-Targeted Therapy

HER2 is a membrane tyrosine kinase, which is part of the epidermal growth factor receptor (EGFR) family. The overexpression of HER2 promotes cell survival, proliferation, and colonization of the CNS [56]. Multiple HER2 tyrosine kinase inhibitors (TKIs) have been investigated in clinical trials demonstrating significant activity in the control of extracranial disease in HER2-positive BC. Trastuzumab is the first HER2 antibody which displayed a remarkable improvement of OS in HER2-positive patients with BC [57], but the large molecular weight (about 148 kDa) precludes the penetration of the intact BBB, leading the CNS to be the most frequent site of relapse [12,58]. The recombinant humanized monoclonal antibody pertuzumab (148 kDa) binds different sites of HER2 receptor, reducing the dimerization of HER2/HER3 receptors and resulting in a double blockage when associated with trastuzumab. The CLEOPATRA trial has shown a major impact on OS from the association of pertuzumab, trastuzumab, and docetaxel (56.5 months, 95%CI 49.3–not reached) compared with the combination of trastuzumab and docetaxel (40.8 months, 95%CI 35.8–48.3) [59]. Therefore, the combination of pertuzumab, trastuzumab, and docetaxel has become the standard first-line treatment in HER2-positive advanced BC. Notably, the trial did not enrolled patients with BM/LM. However, the onset of BM was delayed in the pertuzumab/trastuzumab/docetaxel arm (15.0 months) in comparison with trastuzumab/docetaxel arm (11.9 months, HR 0.58, 95% CI 0.39–0.85, *p* = 0.005) [60]. Thus, we may argue that pertuzumab/trastuzumab/docetaxel represents the first example of chemoprevention for BM in HER2-positive BC [61]. The phase II PATRICIA trial is now evaluating the efficacy of pertuzumab (induction dose of 840 mg followed by 420 mg every three weeks) compared with high-dose trastuzumab (6 mg/kg/weekly) in HER2-positive BM pretreated with RT, while LM represents an exclusion criterion for the enrolment. The first interim analysis after an accrual of 15 patients revealed a response rate of 20% based on RANO criteria with a range of duration of response between 1.4–3.3 months. Six patients discontinued treatment (five for disease progression; one for symptomatic deterioration) and no safety concerns were reported, thus the enrolment is still active [62].

A further evolution is represented by the antibody-drug conjugate trastuzumab emtansine (T-DM1), that has been shown to improve OS in patients with trastuzumab-resistant advanced BC and asymptomatic BM previously treated with RT, compared with lapatinib plus capecitabine [63]. Two additional studies also reported some evidence of activity for T-DM1 in patients with HER2-positive BC and BM [64,65]. The KAMILLA trial is a phase IIIb study of T-DM1 on 398 patients with BM from HER2-positive locally advanced/metastatic BC with prior HER2-targeted therapy and chemotherapy. A partial response according to RECIST v1.1 criteria was achieved in 42.9% of patients (95% CI 34.1–52.0), including 49.3% (95% CI 36.9–61.8) of 67 patients without prior RT to BM. Median PFS and OS were 5.5 (95% CI 5.3–5.6) months and 18.9 (95% CI 17.1–21.3) months, respectively [66]. To date, one case-report only reported a significant clinical and radiological response to a combination of T-DM1 and WBRT lasting > 3 months in patient with HER-2 positive LM heavily pre-treated [67].

New HER2-TKIs with better penetration in the CNS are under investigation. Lapatinib is an orally small molecule, that binds HER2 and EGFR family. A phase II study have investigated lapatinib as single agent in pre-treated patients with BM showing an intracranial response rate of 6% only [68]. However, when lapatinib was associated with capecitabine in a randomized phase II trial, the response rate increased up to 20–38% [69,70,71]. The phase II LANDSCAPE trial has investigated the association of capecitabine plus lapatinib as an up-front treatment in HER2-positive BM, reporting an intracranial response rate of 65.9% (95%CI 50.1–75.9%), and suggesting that such a combined therapy, instead of RT, could be a feasible first-line treatment in HER2-positive BM from BC [72]. The phase III EMILIA trial has compared TDM-1 versus capecitabine plus lapatinib in 991 metastatic HER-2 positive BC following trastuzumab. Patients with BM treated with TDM-1 achieved a longer OS (26.8 months) compared with capecitabine plus lapatinib (12.9 months, HR 0.38; *p* = 0.008) [63,73]. Thus, TDM-1 has been approved as second-line treatment in HER2-positive BC.

Recently, Morikawa et al. have investigated in a phase I trial whether the intermittent high dose of lapatinib (1500 mg twice daily on day 1–3 and day 15–17) alternating with capecitabine (1500 mg twice daily 1500 mg on day 8–14 and day 22–28) might be safe and effective in BM (4 patients), LM (5 patients), and intramedullary metastases (two patients). The study demonstrated a good tolerability. However, the authors have too few patients with measurable disease in each CNS metastasis type to meaningfully examine response rate or PFS. Therefore, a phase II randomized trial (with separate cohorts for BM and LM patients), comparing the standard of care or physician’s choice (anti-HER2 TKIs) with non-TKIs with proven CNS activity, such as TDM-1, will be designed [74].

Temozolomide (TMZ), which represents the standard of care in glioblastoma, has been investigated in association with lapatinib in HER2-enriched BM. The phase I LAPTEM trial has enrolled 16 patients with HER2-positive BM heavily pre-treated with different combination of therapy. The lapatinib-TMZ regimen showed a favourable toxicity profile. A stable disease was achieved in 10/15 (66.7%) patients with a median PFS of 2.6 months (95%CI 1.8–3.34) and a median OS of 10.9 months (95%CI 1.1–20.8) [75]. Recently, Zimmer et al. have shown that low doses of TMZ administered in a prophylactic, metronomic fashion can significantly prevent the development of BM in murine models of BC [76]. Based on these findings, a secondary prevention clinical trial has been designed with oral TMZ given to HER2-positive BC patients with BM following local treatment in combination with T-DM1 for systemic control of disease. The primary endpoint is one-year PFS (NCT03190967).

Neratinib is an oral, irreversible pan-inhibitor of HER family which was investigated in a phase II trial of the Translational Breast Cancer Research Consortium (TBCRC) enrolling pre-treated, symptomatic HER2-positive BM. Neratinib was provided for four different cohorts: (1) neratinib monotherapy; (2) neratinib after surgery; (3a) neratinib plus capecitabine without previous lapatinib; (3b) neratinib plus capecitabine in patients pre-treated with lapatinib. Forty patients treated with neratinib alone achieved an intracranial response rate of 8% (95%CI, 2% to 22%) after a median number of cycles of 2 (1–7 cycles), with a median PFS of 1.9 months [77]. The most common grade ≥ 3 adverse was diarrhea (occurring in 21% of patients taking prespecified loperamide prophylaxis and 28% of those without prophylaxis) associated with a significant worsening of quality of life. Similar to lapatinib, the response rate increased with the association of capecitabine. In fact, the intracranial response rates were of 49% (95%CI 32–66) and 33% (95%CI 10–65) in cohort 3a and 3b, respectively. Interestingly, two patients in the cohort 3a and 1 patient in the cohort 3B presented with LM: one had a partial response after seven cycles, one had stable disease after four cycles of therapy, and one developed progression during cycle 1 [78]. The phase III NALA trial has compared capecitabine plus neratinib (307 patients) versus capecitabine plus lapatinib (314 patients) as third- or later-line therapy. The six- and 12-month PFS rates were of 47.2% versus 37.8% and of 28.8% versus 14.8% for neratinib plus capecitabine versus lapatinib plus capecitabine, respectively. OS rates at six and 12 months were 90.2% versus 87.5% and 72.5% versus 66.7% for neratinib plus capecitabine versus lapatinib plus capecitabine, respectively (HR = 0.88; 95%CI 0.72–1.07; *p* = 0.2086). Intracranial response rate was improved with neratinib plus capecitabine versus lapatinib versus capecitabine (32.8% vs. 26.7%; *p* = 0.1201). Notably, neratinib plus capecitabine postponed time to intervention for symptomatic BM (overall cumulative incidence 22.8% versus 29.2%, *p* = 0.043) [79]. Based on these results, since 2019 both capecitabine plus lapatinib and capecitabine plus neratinib are listed as therapeutic options in NCCN CNS tumor practice guidelines. Focused trials on neratinib in LM are still lacking and represents an unmet need.

Afatinib is an oral, irreversible HER1–2 TKI with a remarkable activity in both BM and LM from NSCLC [80,81]. The phase 2 LUX-Breast 3 trial has compared afatinib alone, afatinib plus vinorelbine, or standard of care per institution (SOC) in 121 HER2-positive BM pre-treated with HER2-targeted agents (trastuzumab, lapatinib, or both). Surprisingly, clinical benefit was higher after SOC (41.09%) and toxicity was higher in patients treated with afatinib regimens [82]. Hence, these disappointing results have discouraged further investigation of afatinib in BC. However, a randomized phase II trial (NCT04158947) will investigate the combination of T-DM1 with afatinib.

Tucatinib is an oral, reversible and selective HER2 TKI with reduced side effects (diarrhea and rash) compared with neratinib and lapatinib. Tucatinib has been studied in association with capecitabine and trastuzumab in a phase I trial on 12 patients with HER-2 positive BM, reporting a significant radiological response in five patients (42%) [83]. The phase II randomized HER2CLIMB trial enrolled patients with HER2-positive advanced BC previously treated with trastuzumab, pertuzumab, and T-DM1 to receive tucatinib or placebo, in combination with trastuzumab and capecitabine. Patients with both asymptomatic and symptomatic BM could be enrolled, while LM was excluded. For patients with BM (*n* = 291), one-year PFS was 24.9% in the tucatinib-combination group and 0% in the placebo-combination group (HR 0.48; 95%CI, 0.34 -0.69; *p* < 0.001), and the median PFS was 7.6 months and 5.4 months, respectively [84]. Tucatinib has received the approval by FDA to be administered in combination with trastuzumab and capecitabine in patients with advanced HER2-positive BC, with or without BM. New combinations of therapy with tucatinib are under investigation. For instance, the HER2CLIMB-02 trial (NCT03975647) will assess the efficacy of tucatinib plus T-DM1 or placebo in patients with BM with the primary endpoint of PFS.

Pyrotinib is an oral, irreversible HER1-2-4 TKI that has displayed in a phase III trial a significant activity in association with capecitabine in terms of PFS (11.1 months) compared with placebo (4.1 months, *p* < 0.001) in HER2-positive BC who were pre-treated with taxane and trastuzumab. However, the enrolment of BM or LM was not allowed [85]. A retrospective cohort of 168 patients with HER2-positive BC revealed a median PFS of 8.8 months (95%CI 6.6–11.0) in patient with BM (*n* = 39) who were heavily pre-treated with HER2-targeted agents [86]. Now, two different phase II trials (NCT03691051 and NCT03933982) are active for the investigation of pyrotinib in BM.

Epertinib (S-222611) is a potent reversible inhibitor of HER2-4 and EGFR. A dose-escalation phase I/II trial has evaluated the tolerability and antitumor activity of epertinib combined with trastuzumab (arm A), with trastuzumab plus vinorelbine (arm B), or with trastuzumab plus capecitabine (arm C), in patients with HER2-positive BC, including with BM. The intracranial response rate was of 67% (*n* = 9) in patients treated with epertinib, 56% (*n* = 9) in patients treated with trastuzumab plus capecitabine, and 0% in those who received trastuzumab plus vinorelbine (*n* = 5) [87].

Other targeted therapies, which do not target the HER family, have been evaluated in HER2-positive BM or LM from BC. Bevacizumab, an anti-VEGF monoclonal antibody, interfere with the permeability of blood vessels and improve the drug delivery to BM. Hence, bevacizumab has been investigated in different phase II trials in association with other antineoplastic treatments. Lin et al. reported in 38 patients with BM treated with bevacizumab associated with carboplatin and trastuzumab an intracranial response rate of 63% compared with 45% after carboplatin and trastuzumab [88]. Lu et al. described a similar efficacy of bevacizumab when associated with etoposide and cisplatin (BEEP regimen) with an intracranial response rate of 77.1% compared with etoposide and cisplatin alone of 54.3% [89]. Interestingly, BEEP regimen in HER2-positive LM has displayed some activity in 19/34 patients (68%) with a median OS of 13.6 months [90].

Cabozantinib is a small, multiple TKI, that binds MET and VEGF receptor 2, and has shown a significant activity in BM from NSCLC. A single-arm phase II study enrolled patients with new or progressive BM into three cohorts: 1) HER2-positive; 2) ER-positive/HER2-negative; 3) TNBC. Although cabozantinib was well tolerated, intracranial response rate was disappointing (5% in cohort 1, 14% in cohort 2, and 0% in cohort 3) [91].

The upregulation of phosphoinositide-3-kinase/mammalian target of rapamycin (PI3K/mTOR) has been described as a mechanism of trastuzumab resistance in HER2-enriched BC [92]. In this regard, the phase II LCCC 1025 trial has investigated whether everolimus in association with vinorelbine and trastuzumab was effective in 32 patients with BM. Intracranial response rate was of 4% only (1 partial response), but the median PFS was 3.9 months (95%CI 2.2–5), and the median OS was 12.2 months (95% CI 0.6–20.2) [93]. Therefore, targeting PI3K/mTOR pathway may represent a further field of research, and ongoing clinical trials are now evaluating different PI3K inhibitors, such as GDC-0084 (NCT03765983), BKM120 (NCT02000882), MEN1611 (NCT03767335), and copanlisib (NCT04108858).

### 5.2. Novel Options for ER Positive BC

Endocrine therapy (ET) is the preferred option for ER-positive BC: tamoxifen or fulvestrant are the first choice of treatment in pre-perimenopausal women, while a Luteinizing Hormone Releasing Hormone (LHRH) agonist is the preferred option for post-menopausal women. The proliferation of ER-positive BC cells is strictly dependent on cyclin D1, which regulates the activity of cyclin-dependent kinase 4 and 6 (CDK 4/6) that promotes tumor invasion. The CDK4/6 inhibitors combined with ET is the standard of care for post-menopausal ER-positive/HER2 negative BC, and in combination with an LHRH agonist for pre-menopausal women, reporting an improvement in PFS, but a limited efficacy in BM control [94,95]. Abemaciclib has a higher BBB penetration compared with other CDK4/6 inhibitors, and therefore has been evaluated in the phase II JBPO trial reporting an intracranial response rate of 5.6% only, and a clinical benefit lasting > 6 months in 25% of patients with BM [96]. Several trials are ongoing to evaluate abemaciclib for BM from BC (NCT02308020, NCT03846583). An open issue is whether the use of CDK4/6 inhibitors in the early stage of ER positive/HER2-negative BC may have an impact on timing of development of BM. Moreover, up to 20% of ER-positive/HER2-negative metastatic BC are intrinsically resistant to CDK4/6 inhibitors, and nearly all patients, whose tumor initially responded to these drugs, will develop acquired resistance. In this regard, new compounds, such as the estrogen receptor degraders (SERDs), with the aim to induce a degradation of estrogen receptors in order to block the ER pathway [97], are under investigation in clinical trials in metastatic BC (NCT02248090, NCT2338349). Overall, the intracranial efficacy of SERDs is still unknown and will be investigated in future clinical trials.

### 5.3. Novel Options for TNBC

BM occur in approximately 50% of patients with TNBC with a median OS <4 months, whose 80% progress with extracranial metastases [98]; thus, the development of effective systemic therapy is an urgent and unmet need. The standard of care in advanced TNBC, regardless of the BRCA (breast cancer susceptibility genes 1 or 2) status, previously treated with anthracyclines with or without taxanes in neoadjuvant or adjuvant setting, is represented by platinum-based chemotherapy, such as carboplatin, which has demonstrated a better tolerability compared with docetaxel [95]. BRCA 1 and 2 (breast cancer susceptibility genes 1 or 2) are suppressor oncogenes involved in repairing DNA double-strand breaks. Mutations in these genes make BC cells unable to repair DNA double strand-breaks through poly adenosine diphosphate ribose polymerase (PARP) enzymes, causing DNA alterations and tumor cell death. The PARP inhibitors iniparib, olaparib, talazoparib, and veliparib have been investigated in metastatic TNBC. The phase II TBCRC trial on iniparib associated with irinotecan has shown modest benefit in 34 patients with BM: intracranial response rate was 12% only, median PFS 2.1 months, and median OS 7.8 months [99]. The phase III EMBRACA trial compared the efficacy and safety of talazoparib with physician’s choice of chemotherapy in 431 patients with TNBC harboring a germline BRCA1/2 mutation and reported an improvement of PFS and objective response rate with PARP inhibitors. Interestingly, the advantage was observed also in 63 patients with asymptomatic BM [100]. The association of carboplatin and ABT888 (veliparib) has been demonstrated to cross the BBB and improve survival in BRCA-mutant intracranial TNBC murine models [101]. Now, an ongoing trial (NCT02595905) is evaluating the combination of carboplatin and veliparib in patients with active BM.

Programmed death ligand 1 (PDL-1) is expressed in approximately 54% of BM from BC [102], and immunotherapy is particularly attractive in TNBC. Recently, the phase III IMpassion 130 trial has evaluated in 902 patients with metastatic TNBC atezolizumab versus placebo in association with nab-paclitaxel. Median PFS and OS were longer in atezolizumab arm (7.2 and 21.3 months, respectively) in comparison with placebo (5.5 and 17.6 months, respectively). Approximately 7% of patients in each arm had BM, but a significant benefit from atezolizumab has not observed [103]. Based on these results, the association of atezolizumab plus nab-paclitaxel has been proposed as an option for first-line therapy for PD-L1 positive advanced TNBC with visceral metastases (Level of Evidence IB according to the 2020 ESO/ESMO Guidelines), but further clinical trials must investigate the efficacy of immune checkpoint inhibitors (ICIs) in patients with CNS recurrence [95]. In fact, some ongoing studies are now assessing the efficacy of ICIs, such as nivolumab (NCT03807765), pembrolizumab (NCT03449238), and atezolizumab (NCT03483012) in combination with SRS in BM from TNBC.

A different therapeutic approach consists of modifying the structure of traditional chemotherapeutic agents using peptide vector or pegylation to improve the penetration through the BBB. ANG1005, a novel taxane agent, consists of 3 paclitaxel molecules covalently linked to Angiopep-2, designed to cross the BBB and blood-cerebrospinal barriers, and to penetrate into malignant cells via LRP1 transport system [104]. Kumthekar et al. have investigated in a phase II study the activity of intravenous ANG1005 at 600 mg/m^2^ every three weeks in 72 patients with BM from BC, of whom 28 with LM. A clinical benefit was reported in 77% of patients, with an intracranial response rate of 15%. Notably, 79% of patients with LM had disease control with a median OS of 8.0 months (95% CI, 5.4–9.4) [105]. Other randomized trials are ongoing to validate this compound in recurrent BM (NCT02048059) and LM (NCT03613181). Etirinotecan pegol (NKTR-102) is a novel long-acting topoisomerase-1 inhibitor designed to improve safety and efficacy of irinotecan by generating lower peak plasma concentrations and a longer half-life of the SN38, the active irinotecan metabolite, from 2 to approximately 38 days. The phase 3 BEACON trial allowed the inclusion of patients with stable BM, which were heavily pre-treated with anthracycline, taxane, and capecitabine, for evaluating the activity of etirinotecan pegol versus standard of care (eribulin, ixabepilone, gemcitabine, vinorelbine, paclitaxel, docetaxel, nab-paclitaxel). The subgroup analysis on 67 patients with BM revealed that etirinotecan pegol confers a prolonged median OS (10.0 months) compared with standard of care (4.8 months, HR 0.51, *p* < 0.01) [106]. The phase III ATTAIN trial is now investigating whether etirinotecan pegol compared to standard of care (eribulin, ixabepilone, vinorelbine, gemcitabine, paclitaxel, docetaxel, or nab-paclitaxel) may be active in 220 patients with BM from TNBC. As the ATTAIN is a registration trial, etirinotecan pegol may become a new therapeutic option for patients with BM if will meet the primary endpoint of OS [107].

Tesetaxel is an oral taxane which is being investigated in the two-cohort phase II CONTESSA TRIO trial (NCT03952325). Cohort 1 involves patients with advanced or metastatic TNBC for receiving tesetaxel or three inhibitors of PDL-1 (nivolumab, pembrolizumab, atezolizumab). Cohort 2 involves elderly patients (≥ 65 years) that receive tesetaxel monotherapy. The trial allows the enrolment of patients with stable BM, but not those with LM.

Some evidence of efficacy of eribulin mesylate has been reported in pre-treated patients with metastatic BC based on results of the phase III 305/EMBRACE study [108]. Recently, Adamo et al. reported in a prospective cohort of 118 patients treated with eribulin mesylate as third-line therapy an objective response rate of 16% (12 patients) in those with BM, with a PFS of 5.2 months (95% CI 2.8–8.4) [109]. A phase II trial (NCT02581839) focused on the activity of eribulin mesylate in BM from TNBC has completed the accrual in July 2020 and results are awaited. Table 1 aims to summarize the main clinical trials with available results on BM from BC.

## 6. Mechanisms of CNS Recurrence and Potential Targets of Treatment

The development of CNS metastases has been suggested to be triggered by a select pool of cells from the primary tumor [110]. Some studies have displayed that primary tumor cells and metastatic cells from BC may share similar genetic alterations. A study on 15 primary BC samples has compared genetic alterations with BM using next generation sequencing and reported no significant difference in the mutational profile. Interestingly, druggable mutations, such as PIK3CA, MLH-1, RB1, and KIT, were found in both primary tumor and BM [111]. Similarly, a study has investigated 19 oncogenes in 12 primary BC and BM displaying that most of mutations were shared between primary tumor and BM, with the exception of EGFR mutation that has been detected in primary BC only [112]. However, Brastianos et have described a molecular divergence in 53% of matched primary tumors and BM from breast, lung and renal cancer with druggable mutations on BM not detected in matched primary tumors [113]. In this regard, approximately 16–22% of ERBB2/HER2 negative BC have been reported to gain ERRB2/HER2 amplifications or somatic mutations in BM, as well as EGFR overexpression has been found in BM, but not in bone metastases [56]. Similarly, while PTEN mutations are rare in primary BC, PTEN loss and mutations have been found in 31% and 21% of BM, respectively [114,115]. Thus, these mutations seem to be exclusive of BC metastatic cells and are crucial for the initial metastatic niche in the brain. Moreover, some of these molecular alterations are not involved in the initial metastatic process, but are important to continue BM growth, suggesting the presence of a metastatic cascade depending on the selection of clonal tumor cells with peculiar ability to survive into the blood stream, to penetrate into the CNS, and survive in the CNS microenvironment. Kienast et al. have described the main steps of the metastatic cascade in BM, which include a dissociation of tumor cells from the primary tumor, invasion of stroma and basal membrane, tumor cell diffusion through the blood circulation, extravasation and penetration through the BBB, and CNS invasion [116].

The role of immune system to promote CNS metastatization is crucial. Mustafa et al. have shown that T lymphocytes may promote the formation of BM in patients with ER negative BC. In particular, the proteomics analysis of the BC cells revealed that guanylate-binding protein 1 (GBP1), a T-cell induced protein, plays a key role in favouring BC to cross the BBB. The GBP1 gene is overexpressed in BC of patients who developed BM, while the silencing of GBP1 reduce the ability of BC cells to cross the in vitro BBB model [117].

Both brain endothelial cells and circulating BC cells are actively involved in the extravasation and penetration through the BBB. The expression of integrin α_v_β_3_ on BC cells increases the VEGF expression on endothelial cells and promotes the neoangiogenesis leading to a reduction of hypoxia-induced apoptosis and the prevention of BC death [118]. Furthermore, integrin α_v_β_3_ increases the BC cells arrest in circulation and interacts with metalloproteinase 9 (MMP-9) with promigratory activity, improved BC cell motility and migration into the CNS [119]. In parallel, the upregulation of VEGF increases both the BBB permeability and the expression of hypoxia-inducible factor (HIF) 2α, ephrin 1 and angiopoietin 2 on endothelial cells promoting BM vascularity [120]. BC cells with CD44+/CD24- have a pro-invasive phenotype due to a significant expression of interleukin (IL) 1α-6-8, urokinase plasminogen activator, MMP-2, neuroserpin and serpin B2 which allow to break basal membrane of the BBB [121,122]. The chemokine stromal cell-derived factor 1α (CXCL12) and the receptor CXCR4 have been found overexpressed in BM from BC and are both involved in migration of BC cells through the BBB [123,124]. MMP-1 expression is positively associated with a high level of COX-2, HBEGF, and SST6GALNAC5 genes, which are identified as BC cells promoters of the passage through the BBB [125,126]. MMP-1 overexpression has a significant role in promoting trans-endothelial migration by disrupting the endothelial junctions. MiR-202-3p, a micro-RNAs that negatively regulates gene expression at the post-transcriptional level, is downregulated in BM compared with primary BC and directly targets MMP-1. Loss of miR-202-3p in BC cells improved the transmigration through BBB by upregulating MMP-1 and disrupting the endothelial junctions, including claudin-5, ZO-1 and ß-catenin. Restoring miR-202-3p expression leads to a metastasis-suppressive effect and preserves the endothelial barrier integrity [127]. Overall, multiple pathways modulate the expression of MMP proteins, which regulate the BBB integrity and promote the trans endothelial migration of tumor cells into the CNS. Another factor involved in colonization of CNS is represented by cathepsine S, which acts as promoter of trans endothelial migration of BC cells by proteolytic effect on junctional adhesion molecule B expressed on stroma cells [128].

Angiopoietin-2 (Ang-2) has been described as a potent vasoactive mediator that contributes to enhance the tumor endothelial cell adhesion and the trans endothelial migration. A high level of Ang2 has been associated with impaired tight junctions of endothelial cells and increased permeability of BBB resulting in a more aggressive colonization from TNBC cells [129].

The astrocytic sphingosine-1 phosphate receptor 3(S1P3) represents a novel mechanism of brain colonisation through the secretion of cytokines, such as IL-6 and CCL2, that increases the permeability of BBB and extravasation of tumor cells [130].

Recently, Tavora et al. reported the critical role of the axon-guidance gene Slit2 in endothelium of mouse models of breast and lung cancer. The study revealed a different expression between the endothelial (high Slit2 expression) and tumoral (low Slit2 expression) compartments. Endothelial-derived SLIT2 protein and its receptor ROBO1 promoted the migration of cancer cells towards endothelial cells and intravasation. Conversely, the deletion of tumoral Slit2 enhanced metastatic progression. Moreover, the authors identified a double-stranded RNA derived from tumor cells, that induced a chemotactic signalling pathway in endothelium driving intravasation and metastasis [131].

Some metabolic changes are adopted by BC cells for adapting to CNS microenvironment. In this regard, some enzymes involved in gluconeogenesis, glycolysis, oxidative phosphorylation, lipolysis are upregulated to meet the high energy demands and survive in a low glucose environment [132,133,134,135].

The neural microenvironment, including astrocytes, microglia and resident macrophages, interacts with tumor cells, and create favourable conditions for survival, colonization and outgrowth of BC cells [132]. Direct contacts between BC cells and astrocytes increase the production of inflammatory cytokines (IFNα and TNF), which activate signal transducer and activator of transcription 1 (STAT1) and NFkB in BC cells, promoting tumor growth [136]. Interestingly, Valiente et al. have shown in a preclinical model of BM, including HER2- positive and TNBC, that a subpopulation of reactive astrocytes with activated STAT3 promotes the metastatic microenvironment in a twofold way: STAT3-positive astrocytes reduce the CD8+ lymphocyte infiltration around the metastatic lesion by secretion of infiltration-suppressive protein, such as VEGF-A and TIMP-1, resulting in an inhibition of the acquired immune response. Furthermore, STAT3+ astrocytes interact with CD74+ microglia resulting in a proliferative signal for BM [137]. Similarly, some Authors have demonstrated that STAT 3 controls the VEGF receptor 2 expression on endothelial cells of BBB and upregulate the mitogen extracellular signal-regulated kinase 5 (MEK5) which promotes BC cell invasion [138,139]. In light of these results, silibilin, a nutraceutical compound able to cross the BBB and inhibit STAT3+ astrocytes, has significantly reduced experimental BM in preclinical models, also in advanced stages of colonization. Thus, these results support the investigation of silibilin supplementation in the clinical setting.

Caveolin-1 acts as a tumor suppressor in BM from BC mimicking the effect of STAT3 activation by reducing the expression of MMP-2 and MMP-9 and the invasive ability of BC cells [140,141].

Pro-inflammatory molecules, such IL1β produced by BC cells, activate the microglia expression of JAG1, which is the ligand of Notch receptor on BC cells, and promote tumor growth [142].

Ren et al. have demonstrated preclinical models of TNBC that a subpopulation of BC stem cells interacts with astrocytes that express high protocadherin 7/β-phospholipase C (PCDH7-PLCβ) levels. The PCDH7-PLCβ complex binds Cx43 and create a junction between BC stem cell and astrocyte leading to a decrease of intracellular Ca^2+^ in tumor cells and facilitate resistance to chemotherapy. In parallel, the activation PLCβ induce secretion of NFKβ and STAT1 resulting in an aggressive colonization of the premetastatic niche. Therefore, edelfosine and meclofenate have been suggested as potential therapeutic strategies to block PLCβ and Cx43 gap junction gaping, respectively [143]. Xing et al. have identified a specific long non-coding RNA (X-inactive-specific transcript-XIST), that is downregulated in BM from BC and increases the secretion of immune suppressive cytokines in microglia, leading to suppressed T-cell proliferation. A low dose of fludarabine has been tested in BM from BC cells of mouse models, revealing either an inhibition of tumor growth or a delayed onset of new BM [144].

Some evidence of the involvement of exosomal system have been reported in the pathogenesis of BM from BC. In this regard, the exosomal-annexin A2 (exo-AnxA2) is overexpressed in BC cells and regulates the activity of the mitogen-activated protein kinase (MAPK), NF-kβ, and STAT3 pathways, as well as the production of IL-6 and TNF-α leading to an increased angiogenesis and BC cells proliferation. In vivo analysis showed that the presence of exo-Anx-A2 depleted exosomes significantly and decreased BM formation from BC [145].

All these findings highlight the crucial role of the BBB and microenvironment in CNS progression from BC, being a potential target of treatment.

## 7. Conclusions

BM and LM are frequent complications in patients with advanced BC with poor survival. The evolution of antineoplastic treatments, including radiation techniques and systemic therapy, has led clinicians to improve the selection of patients using prognostic scales in order to choose the adequate treatment. In general, there is lack of randomized trials focused on CNS recurrence making difficult to support specific treatment strategies. Overall, HER2-targeted therapy allows to achieve significant CNS control in patients with BM, while their efficacy in LM need to be further investigated. CD4/6 inhibitors are widely investigated in ER-positive/HER2-negative BC with remarkable extracranial activity, but their ability to control CNS disease is poorly known. Treatment strategies in CNS recurrence from TNBC represents an urgent unmet need, and immunotherapy, PI3K/mTOR and PARP inhibitors, and newly developed cytotoxic drugs hold promise for the future, as well as targeting the brain microenvironment.

Despite significant advances in treating the CNS disease from BC, some critical issues are still open. For instance, the combination of targeted therapy or immunotherapy with RT, especially SRS, is increasingly adopted, but the risk of radionecrosis and the optimal sequence and timing of RT and systemic therapy remain unclear. The use of targeted agents as first-line treatment in early stage of disease may delay the onset of CNS recurrence or the need of salvage-RT with the advantage to reduce the risk of cognitive impairment and acting as primary and secondary chemoprevention. The molecular divergence of CNS recurrence is a critical point in order to choose an adequate treatment in both CNS disease and primary BC. Thus, the molecular profiling of CNS recurrence should be necessary as a basis of choice. In this regard, the ESO/ESMO Guidelines on advanced breast cancer recommend evaluating the molecular profile of BM in patients who had non-metastatic disease at the diagnosis to confirm the molecular concordance with primary tumor. However, a surgical approach is not always feasible, especially in LM. Therefore, there is a need of alternative techniques to predict molecular subtypes, such as brain imaging or liquid biopsy [146,147], and to monitor effects of antineoplastic treatments. Future clinico-translational trials are warranted to address these issues and validate further effective antineoplastic treatments.

## Figures and Tables

**Table 1 ijms-21-08534-t001:** Clinical Trials on BM in Breast Cancer.

Trials	Number of Patients	Treatment	Results
HER2-therapy			
CLEOPATRA [61] Phase III	106 Arm A: patients Arm B: 51 patients	Arm A: pertuzumab + trastuzumab + docetaxel Arm B: placebo + trastuzumab + docetaxel	OS: Arm A: 56.5 months (95%CI 49.3-not reached) Arm B: 40.8 months (95%CI 35.8–48.3) Median time of development of BM: Arm A: 15.0 months Arm B: 11.9 months
PATRICIA [63] Phase II	15 (interim analysis) Full enrollment planned of 40 patients	Pertuzumab plus highly dose of trastuzumab	Response rate (15 patients): 20% Median duration of response: 1.4–3.3 months No safety concerns
EMILIA [64] Phase III	991 Arm A: 495 patients Arm B: 496 patients 95 Arm A: 45 patients Arm B: 50 patients	Arm A: TDM-1 Arm B: capecitabine plus lapatinib	CNS progression: 9/450 (2.0%) and 3/446 (0.7%) patients without CNS metastases at baseline in the T-DM1 and capecitabine plus lapatinib arms, respectively 10/45 (22.2%) and 8/50 (16.0%) patients with CNS metastases at baseline OS among patients with CNS metastases at baseline: Arm A: 26.8 months Arm B: 12.9 months PFS among patients with CNS metastases at baseline: Arm A: 5.9 months Arm B: 5.7 months
KAMILLA [67] Phase IIIb	398 patients pretreated with HER2-targeted therapy and chemotherapy	TDM-1	Best overall response rate: 21.4% (95%CI 14.6–29.6) Clinical benefit rate: 42.9% (95% CI 34.1–52.0) Reduction in the sum of the major diameters of BM ≥30%: 42.9% (95% CI 34.1–52.0) and 49.3% (95% CI 36.9–61.8) of 67 patients without prior radiotherapy to BM, respectively Median PFS: 5.5 months (95% CI 5.3–5.6) Median OS: 18.9 months (95% CI 17.1–21.3)
Lin et al. 2009 [69] Phase II	242 patients pretreated with trastuzumab and RT	Lapatinib alone	Response rate: 6% Volumetric reduction ≥ 20% of BM: 21%
LANDSCAPE [73] Phase II	45 patients not previously treated with WBRT	Capecitabine plus lapatinib	Response rate: 29/45 patients (65.9%; 95% CI 50.1–79.5)
LAPTEM [76] Phase I	16 patients heavily pretreated with different combination of therapy	Lapatinib plus temozolomide	Response rate: 10/15 patients (66.7%) Median PFS: 2.6 months (95%CI 1.8–3.34) Median OS: 10.9 months (95% CI 1.1–20.8)
TBCRC 022 [78] Phase II	40 patients pretreated and symptomatic BM	Arm 1: neratinib alone Arm 2: neratinib after surgery Arm 3a: neratinib plus capecitabine without previous lapatinib Arm 3B: neratinib plus capecitabine following previous lapatinib	Response rate: Arm 1: 8% Arm 3a: 49% Arm 3b: 33%
NALA [80] Phase III	621 Arm A: 307 patients Arm B: 314 patients	Arm A: capecitabine plus lapatinib Arm B capecitabine plus neratinib	Response rate: Arm A: 26.7% Arm B: 32.8% 6- and 12-months PFS: Arm A: 28.8% and 14.8% Arm B: 47.2% and 37.8% 6- and 12-months OS: Arm A: 72.5% and 66.7% Arm B: 90.2% and 87.5%
LUX-Breast 3 Phase 2 [83]	121 Arm A: 40 patients Arm B: 38 patients Arm C: 43 patients	Arm A: afatinib alone Arm B: afatinib plus vinorelbine Arm C: SOC	Clinical benefit: Arm A: 30.0% (95% CI 16.6–46.5) Arm B: 34.2% (95%CI 19.6–51.4) Arm C: 41.9% (95%CI 27.0–57.9)
HER2CLIMB [85] Phase II	291 patients heavily pretreated	Arm A: tucatinib plus capecitabine plus trastuzumab Arm B: placebo plus capecitabine plus trastuzumab	1-year PFS: Arm A: 24.9% Arm B: 0% Median PFS: Arm A: 7.6 months Arm B: 5.4 months
Macpherson et al., 2019 [88] Phase I/II	45	Arm A: epertinib plus trastuzumab Arm B: trastuzumab plus vinorelbine Arm C: trastuzumab plus capecitabine	Response rate: Arm A: 67% Arm B: 0% Arm C: 56%
Other therapy in HER2-positive BM			
Lin et al., 2013 [89] Phase II	38	Arm A: bevacizumab plus trastuzumab plus carboplatin Arm B: trastuzumab plus carboplatin	Response rate: Arm A: 63% Arm B: 45%
Lu et al., 2015 [90] Phase II	35	Arm A: BEEP regimen Arm B: etoposide plus carboplatin	Response rate: Arm A: 77.1% Arm B: 54.3%
Leone et al., 2020 [92] Phase II	36 Cohort 1: 21 HER2-positive patients Cohort 2: 7 ER positive/HER2 negative patients Cohort 3: 8 TNBC patients	Cabozantinib alone (or in association with trastuzumab in HER2 positive patients)	Response rate: Cohort 1: 5% Cohort 2: 14% Cohort 3: 0%
LCCC 1025 [94] Phase II	32	Everolimus plus vinorelbine plus trastuzumab	Response rate: 4% 3- and 6-months clinical benefit rate: 65% and 27% Median intracranial progression: 3.9 months (95% CI 2.2–5) Median OS: was 12.2 months (95% CI 0.6–20.2)
CD4/CD6 inhibitor in ER-positive BM			
JBPO [97] Phase II	58 Cohort A: ER+, HER2- metastatic BC Cohort B: ER+, HER2+ metastatic breasr cancer Cohort C: ER+ with leptomeningeal metastases Cohort D: brain metastases surgical resection	Abemaciclib as monotherapy or with endocrine therapy or with trastuzumab	Cohort A: intracranial response rate of 5.2% (95% CI 0.0–10.9), and intracranial clinical benefit rate of 24% (95% CI 13.1–35.2). Median OS of 12.5 months (95% CI 9.3–16.4). Cohort B: intracranial response rate of 0% and intracranial clinical benefit rate of 11% (95% CI 0.0–23.0). Median OS of 10.1 months (95% CI 4.2–14.3).
TBCRC 018 [100] Phase II	34	Iniparib plus irinotecan	Median intracranial time to progression: 2.14 months Intracranial response rate: 12%, and median Intracranial clinical benefit rate: 27% Median OS: 7.8 months
TNBM			
IMpassion 130 [104] Phase III	902 (7%of patients with BM)	Arm A: atezolizumab plus nab-paclitaxel Arm B: placebo + nab-paclitaxel	Median PFS and OS were longer in atezolizumab arm (7.2 and 21.3 months, respectively) in comparison with placebo (5.5 and 17.6 months, respectively). Patients with BM did not have a significant benefit from atezolizumab
Kumthecar et al., 2020 [106]	100 72 BM 28 LM	ANG1005 alone	Patient intracranial benefit (at least stable disease): 77% Intracranial response rate: 15% (investigator) or 8% In patients with LM, 79% had intracranial disease control and an estimated median OS of 8.0 months (95% CI, 5.4–9.4).
BEACON [107] Phase III	67	Arm A: erinotecal pegol Arm B: SOC	Median OS: Arm A: 10.0 months Arm B: 4.8 months
VESPRY [110] Observational study	118	Eribulin mesylate	Intracranial response rate: 16% Median PFS: 5.5 months (95% CI 4.2–6.6) Median OS: 31.8 months (95% CI 27.9–34.4)

OS: overall survival; TDM-1: trastuzumab emtansine; PFS: progression-free survival; CNS: Central Nervous System; RT: radiotherapy; WBRT: whole-brain radiotherapy; SOC: standard of care; BEEP: bevacizumab, etoposide, carboplatin; HER2: human epidermal growth factor receptor 2; TNBC: triple negative breast cancer; BM: brain metastases; LM: leptomeningeal metastases.

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
