# Peer review of "Management of Brain and Leptomeningeal Metastases from Breast Cancer"

_ijms, 2020, doi:10.3390/ijms21228534_

Round 1
Reviewer 1 Report
The authors provide a detailed review of the therapy options of BM and LM of breast cancer (BC). Overview over the studies and novel therapy approaches are provided and discussed. The work is valuable for clinicians and clinical researches.
I have following suggestions for the authors:
-Line 53: please provide the abbreviation of WBC
-Line 54: please provide the abbreviation of CSF
-Line 65-69: You write:"The incidence of BM as first site of recurrence in HER2-positive patients receiving adjuvant trastuzumab was higher (2.56%; 95%CI 2.07-3.01) compared with those who did not receive trastuzumab (1.94%; 95%CI 1.54-2.38) with an increased relative risk of 1.35 (95%CI 1.02-1.78, p=0.038) to have a CNS recurrence following adjuvant trastuzumab" ->please discuss. Are there reasons why patients with trastuzumab had a higher rate of BM?
- The novel therapeutic agents for TNBC and ER+ BC are described in a very detailed manner and give a very good overview. I suggest to give additionally a short overview of the standard currently recommended systemic treatment options of BM and TNBC and BM of ER+ BC.
Author Response
The Authors thank the reviewer for the suggestions to improve the manuscript.
The authors provide a detailed review of the therapy options of BM and LM of breast cancer (BC). Overview over the studies and novel therapy approaches are provided and discussed. The work is valuable for clinicians and clinical researches.
I have following suggestions for the authors:
R1. Line 63: please provide the abbreviation of WBC.
A1: We have added white blood cells (WBC) in the text
R2. Line 64: please provide the abbreviation of CSF
A2. We have inserted cerebrospinal fluid (CSF) in the text
R3. Line 65-69: You write: "The incidence of BM as first site of recurrence in HER2-positive patients receiving adjuvant trastuzumab was higher (2.56%; 95%CI 2.07-3.01) compared with those who did not receive trastuzumab (1.94%; 95%CI 1.54-2.38) with an increased relative risk of 1.35 (95%CI 1.02-1.78, p=0.038) to have a CNS recurrence following adjuvant trastuzumab" ->please discuss. Are there reasons why patients with trastuzumab had a higher rate of BM?
A3. We have explained in the text the main reason of higher rate of BM following trastuzumab (see lines 74-77)
R4. The novel therapeutic agents for TNBC and ER+ BC are described in a very detailed manner and give a very good overview. I suggest to give additionally a short overview of the standard currently recommended systemic treatment options of BM and TNBC and BM of ER+ BC.
A4. We have reported in the text a brief summary of standard recommended systemic therapy for advanced TNBC and ER+ BC (see lines 456-462, 479-482, and 503-507)

Reviewer 2 Report
The review tries to describe brain metastases in breast cancer patients and the treatments that are being implemented with a wealth of numbers and percentages. Several topics that include the locoregional therapy (WBRT, SRS, Intrathecal therapy) and systemic chemotherapy/ targeted chemotherapy have been reviewed. They have also given a small paragraph to the description of targeting microenvironment as emerging strategy. Few opinions are advised before publication.
1. The title “selection criteria for antineoplastic treatment” is inadequate in describing the paragraph (line 70-118), which was more likely reviewing the prognostic scale.
2. For the locoregional treatment in breast cancer. The description on neurosurgical tumor resection was missing, which was the first line therapy in some breast cancer related BM (also some of the LM) patients.
3. The data implementing the OS/PFS regarding BM to WBRT/SRS include all BM of non-specific origin. I would expect more from subgroup analysis base on breast cancer patients, also further detailing the outcome of different molecular subtype of breast cancer (ER/PR/HER2), and the underlying mechanisms.
4. Propensity to LM from different molecular subtype was evident, however is missing. For the intrathecal therapy, the authors have mentioned the efficacy of intrathecal targeted therapy in HER2 patients. Also I would expect more from the other subtype.
5. The description of targeting brain microenvironment in breast cancer should be the most emphasized, regarding the title of this review and to fit the aims of this special issue: to provide an updated overview of pre-clinical and clinical knowledge on the pathophysiology and molecular profiling of breast cancer), which are missing.
6. A table summarizing the mentioned clinical trials in all different molecular subtype should be included.
7. The authors have mentioned the divergence of CNS recurrence compared with primary solid tumor in the conclusion part, which I think should be elaborated in the review.
A review should not only provide the wealth of numbers and histories, but critical opinion with explanation.Author Response
The Authors thank the reviewer for the suggestions to improve the manuscript.
The review tries to describe brain metastases in breast cancer patients and the treatments that are being implemented with a wealth of numbers and percentages. Several topics that include the locoregional therapy (WBRT, SRS, Intrathecal therapy) and systemic chemotherapy/ targeted chemotherapy have been reviewed. They have also given a small paragraph to the description of targeting microenvironment as emerging strategy. Few opinions are advised before publication.
R1. The title “selection criteria for antineoplastic treatment” is inadequate in describing the paragraph (line 70-118), which was more likely reviewing the prognostic scale.
A1. We have changed the title as following “Prognostic scales in advanced breast cancer with CNS disease” (see line 83)
R2. For the locoregional treatment in breast cancer. The description on neurosurgical tumor resection was missing, which was the first line therapy in some breast cancer related BM (also some of the LM) patients.
A2. We have added more details regarding the role of surgery in both BM and LM (see lines 134-140)
R3. The data implementing the OS/PFS regarding BM to WBRT/SRS include all BM of non-specific origin. I would expect more from subgroup analysis base on breast cancer patients, also further detailing the outcome of different molecular subtype of breast cancer (ER/PR/HER2), and the underlying mechanisms.
A3. More details have been provided in the text (see lines 182-193)
R4. Propensity to LM from different molecular subtype was evident, however is missing. For the intrathecal therapy, the authors have mentioned the efficacy of intrathecal targeted therapy in HER2 patients. Also I would expect more from the other subtype.
A4. We have inserted in the text more details on the efficacy of intrathecal therapy in HER2 negative patients (see lines 285-291)
R5. The description of targeting brain microenvironment in breast cancer should be the most emphasized, regarding the title of this review and to fit the aims of this special issue: to provide an updated overview of pre-clinical and clinical knowledge on the pathophysiology and molecular profiling of breast cancer), which are missing.
A5: we have added more preclinical data and information on the pathophysiology of the development of CNS recurrence (see lines 569-660)
R6. A table summarizing the mentioned clinical trials in all different molecular subtype should be included.
A6: done
R7. The authors have mentioned the divergence of CNS recurrence compared with primary solid tumor in the conclusion part, which I think should be elaborated in the review.
A7: the divergence of CNS recurrence of BM has been developed accordingly (see lines 545-561)
